# Evaluating Inflorescence Morphology in Two Species and Subspecies of the Genus *Hierochloë* R. Brown

**DOI:** 10.3390/plants14152270

**Published:** 2025-07-23

**Authors:** Károly Penksza, Tünde Szabó-Szöllösi, András Neményi, László Sipos, Szilárd Szentes, Zsombor Wagenhoffer, Balázs Palla, Dániel Ákos Balogh, Eszter Saláta-Falusi

**Affiliations:** 1Department of Botany, Institute of Agronomy, Hungarian University of Agriculture and Life Sciences, Páter Károly Str. 1, 2100 Gödöllő, Hungary; penksza.karoly@uni-mate.hu (K.P.); palla.balazs@uni-mate.hu (B.P.); balogh.daniel.akos@phd.uni-mate.hu (D.Á.B.); salata-falusi.eszter@uni-mate.hu (E.S.-F.); 2ELTE Botanical Garden, Illés Str. 25, 1083 Budapest, Hungary; 3Institute of Landscape Architecture, Urban Planning and Garden Art, Hungarian University of Agriculture and Life Sciences, Páter Károly Str. 1, 2100 Gödöllő, Hungary; 4Department of Postharvest Science, Trade, Suppy Chain and Sensory Analysis, Institute of Food Science and Technology, Hungarian University of Agriculture and Life Sciences, Villányi Str. 29-43, 1118 Budapest, Hungary; sipos.laszlo@uni-mate.hu; 5HUN-REN Institute of Economics, Centre for Economic and Regional Studies (HUN-REN KRTK), Tóth Kálmán Str. 4, 1097 Budapest, Hungary; 6Institute for Animal Breeding, Nutrition and Laboratory Animal Science, University of Veterinary Medicine Budapest, István Str. 2, 1078 Budapest, Hungary; wagenhoffer.zsombor@univet.hu

**Keywords:** *Poaceae*, inflorescence morphology, morphometric analysis, taxonomic delimitation

## Abstract

(1) The primary objective was to determine whether, within this taxonomically challenging group, the closely related European species and their subspecies exhibit distinct inflorescence characters that allow for unambiguous differentiation. This study focuses on two closely related species within the genus *Hierochloë*: *Hierochloë hirta* (Schrank) Borbás and *Hierochloë odorata* (L.) Beauv. (2) For four subspecies, data were collected from 15 inflorescences each, while for one subspecies, 10 inflorescences were examined. From each inflorescence, six spikelets were selected. The statistical analyses were non-parametric methods, the Kruskal–Wallis test, and principal component analysis. (3) Morphological traits showed consistent patterns within each subspecies, indicating their suitability for taxonomic differentiation. The most reliable diagnostic traits were the length of the outer glume of the first flower and the lengths of the awns. (4) The study concludes that while some subspecies can be clearly distinguished based on inflorescence morphology, no single trait is sufficient to completely separate all taxa. The authors recommend recognizing *Hierochloë odorata* subsp. *praetermissa* as a subspecies rather than a distinct species and affirm the validity of the species names *Hierochloë hirta* and *Hierochloë odorata*.

## 1. Introduction

In modern taxonomy, molecular analyses are generally considered the most reliable tools for distinguishing among taxa. However, such approaches presuppose the existence of well-defined species, which are frequently represented only by herbarium specimens. This raises a critical issue: how reliably can these preserved specimens be distinguished from one another based on morphological features alone? This study addresses that question by focusing on inflorescence traits, which are commonly used for taxonomic identification in Poaceae family. We selected two closely related European species of the genus *Hierochloë, H. odorata* (L.) Beauv. and *H. hirta* (Schrank) Borbás, along with their subspecies, to assess whether such characters are robust and consistent enough to support taxonomic delimitation when observed in herbarium material.

In addition, the genus is widespread not only in the northern hemisphere but also in the southern hemisphere [1,2]. This taxonomic approach has been supported in various subsequent works, such as the Flora of China [3] and the Flora of North America [4]. In contrast, differing perspectives are observed in the flora of New Zealand and the British Isles [5,6,7,8], while Allred and Barkworth [9] also include North America. Studies by Connor and Renvoize [10] examined the floral biology of Australian and South American *Hierochloë* species without merging the two genera. Species of the genus *Hierochloë* have been described under different names by various authors, so the synonymy is very rich. The genus is also difficult to distinguish from the ideas of other authors in this field and is complex [11,12,13,14].

There are authors who place the genus *Hierochloë* in the genus *Anthoxanthum*. This was recorded by Schouten and Veldkamp [15]. However, this proposal remains controversial. Some authors [8,9,16] have adopted Schouten and Veldkamp’s [15] suggestion. As a result, the species names employed in floristic works include *Anthoxanthum hirtum* (Schrank) Y. Schouten & Veldkamp and *Anthoxanthum nitens* (Weber) Y. Schouten & Veldkamp [14,17]. Chepinoga et al. [18] classified twelve *Hierochloë* taxa within the genus *Anthoxanthum*, including several endemic species. Additionally, they introduced new combinations and elevated subspecies to species rank, such as *Hierochloë odorata* (L.) P. Beauv. subsp. *praetermissa* G. Weim., renamed as *Anthoxanthum praetermissum* (G. Weim.) Chepinoga, comb. & stat. nov. [18].

Certain authors maintain the separation of the *Hierochloë* genus from *Anthoxanthum*, citing differences in floral morphology and chromosome numbers (x = 5 in the *Anthoxanthum* section vs. x = 7 in *Hierochloë*), which are deemed sufficient for recognizing them as distinct genera [12,19,20,21,22]. Despite this, their taxonomic organization remains complex due to the close relationships among species, their morphological similarities, and their cohabitation in shared habitats [23]. Furthermore, members of the *Anthoxanthinae* subtribe are part of multiple polyploid complexes [24,25], which Lema-Suárez [26] clearly demonstrated in their extensive work as supporting evidence for the distinctiveness of the two genera.

Pimentel et al. [25] delineated two sections within the *Anthoxanthum* genus. The *Anthoxanthum* section possesses a hermaphroditic terminal floret and two sterile florets, while the *Ataxia* section contains a hermaphroditic terminal floret and two lower florets, which are mostly male or sterile, often with a palea. Both sections lack lodicules. In contrast, the *Hierochloë* genus is characterized by a terminal floret that is either hermaphroditic or female, accompanied by two male florets, each with lodicules. Lema-Suárez [2,26] and Villalobos et al. [27] clearly identified three distinct sections. The two *Hierochloë* species analyzed in the present study, along with the spikelet floral structures, are visually presented by Conert [28,29], who detailed their spikelets.

However, significant differences in their panicle structures and spikelet morphology (Figure 1a,b) highlight their distinctiveness, despite both genera sharing the characteristics of three-flowered spikelets, where the first two flowers are typically staminate [1].

The proximity of *Hierochloë* and *Anthoxanthum* has led various authors to recognize their close relationship [10]. The Aotearoa/New Zealand taxa were not placed in the genus *Hierochloë*, but in *Anthoxanthum*, due to lack of formal combinations [7,10,30,31,32].

Molecular analyses have revealed that while *Hierochloë* and *Anthoxanthum* exhibit distinct morphological traits, they share significant genetic similarities, complicating their taxonomic classification [33]. *Hierochloë odorata* belongs to the subfamily Pooideae, which encompasses a diverse array of grass species. Phylogenetic studies utilizing molecular markers, such as the internal transcribed spacer (ITS) regions of ribosomal DNA, have greatly advanced our understanding of relationships among *Hierochloë* species. These analyses indicate that *Hierochloë* species, including *Hierochloë odorata* and *Hierochloë hirta*, form a monophyletic group, signifying a common ancestor and highlighting their genetic coherence [34]. This monophyly is critical for understanding the genus’s evolutionary dynamics and its adaptations to specific ecological niches. The taxonomic complexity of *Hierochloë odorata* is further accentuated by its close relationship with other species within the genus, primarily *Hierochloë hirta* and *Hierochloë australis*. These species share overlapping morphological and genetic traits, making the delineation of clear taxonomic boundaries challenging. Recent studies have proposed new taxonomic arrangements based on an integrative approach combining morphological, molecular, and cytological data, clarifying the relationships among these species [34].

Moreover, the two *Hierochloë* species exhibit overlapping distributions in terms of both habitats and geographic ranges, as both are circumpolar in distribution [26]. Beyond its ecological and cultural significance, the phylogenetic relationships of *Hierochloë odorata* can also be interpreted within broader biogeographical contexts. Studies have shown that the distribution of *Hierochloë* species has been influenced by landscape historical factors such as glacial cycles and climatic changes, which have shaped their evolutionary trajectories [26]. The genetic diversity observed within *Hierochloë odorata* populations can be attributed to these landscape historical events, which have facilitated the species’ adaptation to changing environmental conditions.

Comprehensive studies have employed micro- and macromorphological, molecular, and karyological data to clarify their taxonomy, often based on herbarium specimens and increasingly supported by expanding cytological and DNA sequence databases. Former phylogenetic inferences indicated that *Hierochloë* and *Anthoxanthum* are sister taxa encompassing the subtribe *Anthoxanthiae* [25,26]. More recent genetic evidence supports the merging of the two genera, highlighting the intermediate floral structures of *Anthoxathum* section *Ataxia* as a connection between *Anthoxathum* section *Anthoxathum* and *Hierochloë*, in case a higher sampling of *Hierochloë* would be involved [2,26,27]. However, *Hierochloë* forms a well-supported clade within the taxa of the formerly recognized subtribe *Anthoxanthiae*. Phylogenetic evidence revealed the two species as polytomic or sister taxa, occasionally with unresolved support, depending upon the origin of the studied barcoding marker (nuclear or plastid DNA) [35,36,37]. Molecular studies are important because environmental factors can significantly alter the morphological properties of plant individuals [38,39,40,41].

Studies from the Carpathian Basin (in Hungary) have confirmed and clarified that *Hierochloë hirta* is not found in this region, contrary to Borbás’s original assertion [42]. According to floristic works from this area [43,44,45,46,47,48], two species are present: *Hierochloë australis* (Schrad.) Roem. & Schult. 1817, and *Hierochloë odorata* (L.) P. Beauv. 1812. Simonkai [49] reported two species from Transylvania: *Hierochloë australis* and *Hierochloë repens* (Host) Simonk. Somlyay et al. [50] questioned the presence of *Hierochloë odorata* in Hungary. Penksza et al. [51] further demonstrated through morphological measurements that *Hierochloë repens,* not *Hierochloë odorata,* is present in Hungary and Romania [52]. Their results indicated that *Hierochloë odorata* and *Hierochloë repens* significantly differ in the number of spikelets in the inflorescence and the length of the glumes [51]. Notable differences were also observed in the pubescence of the glumes and the length of the awns. Epidermal differences were also characteristic [51].

In his work, Weimarck [53] emphasized two morphological differences between *Hierochloë hirta* and *Hierochloë odorata* that are often difficult to identify. One was the pubescence, as the glumes of the bisexual spikelets of *Hierochloë odorata* bear appressed or only slightly spreading hairs, whereas the glumes of *Hierochloë hirta*’s bisexual spikelets bear distinctly spreading hairs. The other morphological trait was the presence or absence of the awn. The staminate flowers of *Hierochloë odorata* are awnless, with pointed, awned tips or very thin awns, while the male flowers of *Hierochloë hirta* have long awns, rarely pointed, with awned tips.

According to Weimarck [53,54,55,56], the distribution area of *Hierochloë hirta* includes Scandinavia, Western Russia, Poland, the former Czechoslovakia, and Southern Germany. The ecological role of *Hierochloë odorata* extends beyond its habitat preferences; its contribution to biodiversity is well-recognized. As an indicator species for wetland ecosystem health, its presence is often associated with other plant species sharing similar habitat requirements. For example, *Hierochloë odorata* is legally protected in Poland, emphasizing the need for its conservation amidst threats posed by habitat loss and environmental changes [25].

The morphological delineation of species and their subspecies is challenging due to the inherent morphological plasticity of these taxa. Additionally, the genus exhibits limited sexual reproduction, with facultative or obligate apomixis being prevalent. Weimarck’s species and subspecies descriptions have been expanded upon by several researchers. In his descriptions, Weimarck emphasized the presence or absence of the awn and the indumentum of the spikelet components, the latter being a trait that is difficult to quantify.

For European taxa, Wallnöfer [57] accepted the five groups proposed by Weimarck, also considering Conert’s [28] work. These groups include the following: *Hierochloë repens* (Host) P. Beauv. (tetraploid, 2n = 28), *Hierochloë odorata* (L.) P. Beauv. subsp. *odorata* (tetraploid and hexaploid, 2n = 28 and 42), *Hierochloë odorata* (L.) P. Beauv. subsp. *baltica* Gweim. (hexaploid, 2n = 42), *Hierochloë hirta* (Schrank) Borbás subsp. *hirta* (octoploid, 2n = 56), *Hierochloë hirta* (Schrank) Borbás subsp. *arctica* (J. Presl in C. Presl) Gweim. (octoploid, 2n = 56). Wallnöfer conducted extensive field and herbarium research on *Hierochloë repens*, successfully distinguishing this taxon and mapping its European distribution [58,59,60]. He also noted its occurrence in Hungary and Romania, which was confirmed by Penksza and Ruprecht [52]. Furthermore, Perić et al. [60]. reported the species in Serbia, and it has since appeared as an invasive species in Poland [61]. Additionally, Wallnöfer refined the descriptions and distribution data of *Hierochloë odorata* taxa.

The most detailed descriptions of *Hierochloë hirta* and *Hierochloë odorata*, along with their subspecies, were provided by Weimarck [53,56]. He highlighted the characters distinguishing the two species, offering comprehensive descriptions of the species and subspecies, and emphasizing the key differentiating traits. He described two subspecies for each species.

In *Hierochloë odorata*, the lemma of the male flowers has flat or slightly protruding hairs; the lemma is glabrous, membranous, or has very thin hairs with sparse marginal hairs. In *Hierochloë hirta*, the lemma has distinctly protruding hairs. In *Hierochloë odorata* subsp. *odorata*, the lemma almost always ends in a pointed awn. In *Hierochloë odorata* subsp. *baltica*, the lemma is glabrous, rarely with very short hairs. *Hierochloë hirta* subsp. *hirta* has a slightly curved awn; the panicle usually has nine or more nodes and is dark purple-brown when mature; the lower main branches of the panicle are usually pendulous. *Hierochloë hirta* subsp. *arctica* has a straight or finely inward-curved awn; the panicle has up to eight nodes and is golden-brown when mature; the lower main branches of the panicle are usually not pendulous.

Weimarck [53] also provided detailed characterizations of the subspecies, including measurements of the panicle and spikelet. From these descriptions, the characters applied in the present work are as follows:

*Hierochloë odorata* subsp. *odorata* is characterized by a panicle measuring between 35 and 90 mm in length, occasionally reaching up to 125 mm, with typically six to nine nodes, though sometimes as few as four or as many as eleven. The glumes are 3.5 to 5.5 mm long, occasionally ranging from 2.5 to 7.5 mm, glabrous, and exhibit a golden sheen when fully developed. The lower glumes of the staminate flowers measure between 3.0 and 4.5 mm in length, sometimes as short as 2.5 mm or as long as 5.5 mm, are acute and mucronate, adorned with hairs up to 0.3 mm long towards the apex. The hyaline portion at the apex is typically 0.1 to 0.3 mm wide. The margins are sparsely ciliate, featuring simple and occasionally branched hairs that are straight or slightly curly, measuring 0.3 to 0.6 mm in length, occasionally up to 0.8 mm. If present, the awn on the lower glumes of the staminate flowers ranges from 0.1 to 0.5 mm in length, is slightly rough and situated 0.1 to 0.2 mm below the apex. In the upper staminate flowers, if an awn is present, it measures between 0.1 and 0.2 mm, occasionally up to 0.8 mm, and is similarly slightly rough.

In contrast, *Hierochloë odorata* subsp. *baltica* features a panicle ranging from 45 to 75 mm in length, occasionally as short as 30 mm or as long as 100 mm. The glumes are 4.6 to 6.0 mm long, sometimes between 4.0 and 8.0 mm, glabrous, and display a golden sheen upon full development. The lower glumes of the staminate flowers measure between 3.7 and 5.2 mm in length, occasionally from 3.2 to 6.0 mm, are neither acute nor mucronate, but rather finely notched or truncate, or sometimes cleft at the apex, with hairs up to 0.3 mm long towards the apex. The hyaline portion at the apex is typically 0.3 to 0.5 mm wide. The margins are sparsely ciliate, bearing simple or occasionally branched, straight (not curly) hairs measuring 0.5 to 0.8 mm in length. Awns are absent on all lower glumes of the staminate flowers, or if present, are less than 0.1 mm long.

*Hierochloë hirta* subsp. *hirta* is distinguished by a panicle measuring between 75 and 150 mm in length, occasionally as short as 50 mm. The glumes are 4.0 to 5.6 mm long, sometimes up to 6.3 mm, glabrous, and exhibit a golden sheen when fully developed, almost always displaying a distinctly reddish hue. The lower glumes of the staminate flowers measure between 3.0 and 5.0 mm in length, are finely notched or truncate, or sometimes cleft at the apex, with hairs up to 0.5 mm long towards the apex. The margins are densely ciliate, featuring simple and branched straight hairs measuring 0.5 to 1.0 mm in length. The lower glumes of the staminate flowers bear an awn ranging from 0.2 to 0.8 mm in length, slightly rough, originating at the apex or at the base of the apical notch, or if present, 0.1 to 0.5 mm below the apex. In the upper staminate flowers, the awn measures between 0.2 and 1.0 mm in length, occasionally absent, and rough. Both awns taper towards the end and are weakly but distinctly curved outward.

Lastly, *Hierochloë hirta* subsp. *arctica* presents a panicle measuring between 45 and 85 mm in length, occasionally as short as 30 mm. The glumes are 4.5 to 6.0 mm long, sometimes between 4.0 and 6.3 mm, glabrous, and display a golden sheen upon full development. The lower glumes of the staminate flowers measure between 3.5 and 5.5 mm in length, are acute or finely notched, truncate, or sometimes cleft at the apex, with hairs up to 0.8 mm long towards the apex. The awn on all lower glumes of the staminate flowers, if present, ranges from 0.2 to 0.7 mm in length, is rough, straight, or less commonly slightly curved outward or inward, and approximately uniform in width. In the upper staminate flowers, if an awn is present, it measures between 0.1 and 0.2 mm, occasionally up to 0.8 mm, and is slightly rough.

Wallnöfer [57] added to it and drew attention to important features, but these related mainly to the hairiness of the maker of the garland. Based on Weimarck’s [53,56] descriptions, Wallnöfer [57] also placed the *Hierochloë hirta* taxa under *Hierochloë odorata*. The subspecies *Hierochloë hirta* (Schrank) Borbás ssp. *praetermissa* Gweim. (hexaploid, 2n = 42) was also evaluated as a subspecies of *Hierochloë odorata*. The plant prefers anthropogenically influenced habitats, and it is mainly found in Russia. Weimarck [53] suggests that it was probably introduced into Scandinavia, most likely during the Iron Age. However, it is very scattered in central Europe. This includes two populations from the Havel River, populations from the Havel bank near Berlin and populations from Lac de Tanay in Valais, Switzerland [56,62].

Vegetative structures generally provide less reliable diagnostic characteristics for species identification compared to generative traits. In this study, we focused on precisely measurable generative characters within a taxonomic group where character overlap often complicates species delimitation. Our primary aim was to assess whether the examination of the most defining generative traits is sufficient for the unequivocal separation of species. Additionally, we sought to determine whether these characters are appropriate for taxonomic differentiation and whether a single inflorescence trait can effectively distinguish all recognized subspecies within this group.

## 2. Results

In the principal component analysis (PCA), we considered the first two principal components, F1 and F2, which accounted for 77% and 19.13% of the variance, respectively, culminating in a cumulative variance of 96.13%. The characters most strongly correlated with these newly derived components were as follows: for F1, the length of the lower glume (0.980), the length of the spikelet (0.954), the length of the upper glume (0.945), the length of the lemma (0.868), and the length of the palea of the second flower (0.868); for F2, the length of the awn of the lemma of the second flower (0.963) and the length of the awn of the lemma of the first flower (0.960).

Upon analyzing the data through PCA, considering all measured characters (17 inflorescence traits) across the five taxa to assess similarities and differences among them, the 95% confidence ellipses revealed distinct separation of *Hierochloë odorata* subsp. *odorata* from both *Hierochloë hirta* subsp. *hirta* and *Hierochloë odorata* subsp. *baltica*, as illustrated in (Figure 2).

Notably, *Hierochloë hirta* subsp. *arctica* exhibited significant overlap with *Hierochloë odorata* subsp. *baltica*. Additionally, the data for *Hierochloë hirta* subsp. *praetermissa* showed considerable overlap with both *Hierochloë odorata* subsp. *odorata* and *Hierochloë hirta* subsp. *arctica*. The confidence ellipses of *Hierochloë hirta* subsp. *hirta* and *Hierochloë hirta* subsp. *arctica* also overlapped substantially.

Regarding panicle length, differentiation was observed among three subspecies (Table 1). *Hierochloë odorata* subsp. *baltica*, possessing the shortest average panicle length of 59.8 mm, was distinct from *Hierochloë odorata* subsp. *odorata*, which had the longest average panicle length of 80.1 mm, and from *Hierochloë hirta* subsp. *hirta*, with an average length of 80.6 mm. The remaining subspecies did not exhibit significant differences in panicle length.

In terms of the length of the first internode of the panicle, only *Hierochloë odorata* subsp. *baltica*, with an average length of 17.66 mm, differed significantly from *Hierochloë hirta* subsp. *hirta* and *Hierochloë odorata* subsp. *odorata*, which had average lengths of 23.80 mm and 24.33 mm, respectively. No other significant differences were found.

Measurements of the lengths of the longest and second longest panicle branches at the first, second, and third nodes did not reveal significant differences among any of the taxa.

The length of spikelets is a crucial inflorescence character, consistently serving as an important characteristic in identification keys. According to current data, *Hierochloë odorata* subsp. *odorata* possesses the shortest average spikelet length at 4.7 mm. This measurement does not significantly differ from that of *Hierochloë hirta* subsp. *praetermissa*, which averages 5.0 mm. However, it is significantly distinct from the spikelet lengths of the other three analyzed subspecies. *Hierochloë hirta* subsp. *praetermissa* does not show a significant difference from *Hierochloë hirta* subsp. *hirta*, which has an average spikelet length of 5.20 mm. The spikelet measurements of *Hierochloë odorata* subsp. *baltica*, averaging 5.53 mm, are not significantly different from those of *Hierochloë hirta* subsp. *arctica*, which average 5.51 mm.

The glumes surrounding the spikelets are also key features in the identification of grasses. In the examined taxa, it is characteristic for all subspecies that the lower glume is shorter and the upper glume is longer. Based on the lower glume measurements, *Hierochloë odorata* subsp. *odorata* has the shortest average length at 4.31 mm and is significantly distinct from the lower glume lengths of all other analyzed subspecies. *Hierochloë hirta* subsp. *praetermissa* (4.59 mm) and *Hierochloë hirta* subsp. *hirta* (4.71 mm) form a group where their lower glume lengths are not distinguishable. Another group comprises *Hierochloë odorata* subsp. *baltica* (5.08 mm) and *Hierochloë hirta* subsp. *arctica* (5.01 mm), which have the longest measurements.

Regarding the upper glume, the groups identified based on the lower glume are similarly formed. *Hierochloë odorata* subsp. *odorata* has the shortest average length at 4.51 mm and is significantly distinct from the upper glume lengths of all other subspecies, though all other average lengths are greater.

The length of the lemma demonstrated the greatest potential for differentiation (Figure 3). The first flower’s lemma in *Hierochloë odorata* subsp. *odorata* has the shortest average length at 3.87 mm and is significantly distinct from the first flower’s lemma lengths of all other subspecies. The measurements for *Hierochloë hirta* subsp. *praetermissa* (4.10 mm) and *Hierochloë odorata* subsp. *baltica* (4.79 mm) are also significantly distinct from those of all other subspecies. *Hierochloë hirta* subsp. *hirta* (4.54 mm) and *Hierochloë hirta* subsp. *arctica* (4.45 mm) do not show significant differences in their lower glume lengths.

The awn length of the first flower’s lemma serves as a distinguishing characteristic, potentially differentiating *Hierochloë odorata* from *Hierochloë hirta*. Our findings partially support this distinction. *Hierochloë odorata* subsp. *odorata* exhibits a short awn measuring approximately 0.05 mm. Similarly, *Hierochloë hirta* subsp. *praetermissa* possesses a short awn of about 0.025 mm, while *Hierochloë odorata* subsp. *baltica* has the shortest awn among the taxa studied, at 0.002 mm. These three taxa can thus be grouped together. *Hierochloë hirta* subsp. *praetermissa* significantly differs from the other two *Hierochloë hirta* subspecies but not from the *Hierochloë odorata* subspecies. *Hierochloë hirta* subsp. *hirta* has the longest awn, measuring 0.59 mm, and is not significantly different from *Hierochloë hirta* subsp. *arctica*, which also has a long awn of 0.47 mm.

In analyzing the length of the first flower’s palea, it was observed that *Hierochloë hirta* subsp. *praetermissa*, with an average length of 3.62 mm, does not significantly differ from *Hierochloë odorata* subsp. *odorata*, which has an average length of 3.52 mm. Notably, *Hierochloë odorata* subsp. *baltica* exhibits distinctively longer palea measurements, averaging 4.09 mm and thereby setting it apart from the other subspecies.

The length of the second flower’s lemma follows a pattern similar to that of the first flower. *Hierochloë odorata* subsp. *odorata* again presents the shortest average length at 3.68 mm, though this measurement does not significantly distinguish it from *Hierochloë hirta* subsp. *praetermissa*, which averages 3.96 mm. The remaining subspecies display longer lemma lengths, consistent with observations from the first flower.

The awn length of the second flower’s lemma proves to be a valuable distinguishing characteristic (Figure 4). *Hierochloë odorata* subsp. *odorata* possesses an exceptionally short awn, averaging 0.014 mm, while *Hierochloë hirta* subsp. *praetermissa* also features a relatively short awn at 0.039 mm. In contrast, *Hierochloë odorata* subsp. *baltica* is characterized by the absence of an awn. These three taxa thus form a distinct group based on this trait. *Hierochloë hirta* subsp. *hirta* exhibits the longest awn, measuring 0.62 mm, which significantly differentiates it from *Hierochloë hirta* subsp. *arctica*, whose awn measures 0.47 mm.

The length of the second flower’s palea mirrors the pattern observed in the first flower. *Hierochloë hirta* subsp. *praetermissa*, with an average length of 3.61 mm, does not significantly differ from *Hierochloë odorata* subsp. *odorata*, which averages 3.51 mm. Once again, *Hierochloë odorata* subsp. *baltica* stands out with notably longer palea measurements, averaging 4.16 mm, thereby distinguishing it from the other subspecies.

## 3. Discussion

The length of the panicle serves as a crucial identification characteristic, although among the examined taxa, it often exhibits extreme size variation [24,27,28,63,64]. This trait represents the initial step in specimen determination. In the present analysis, differentiation among three subspecies was successfully demonstrated based on this characteristic. *Hierochloë odorata* subsp. *baltica*, which exhibited the shortest average panicle length (59.8 mm), was distinguished from *Hierochloë odorata* subsp. *odorata,* which had the longest average panicle length (80.1 mm), as well as from *Hierochloë hirta* subsp. *hirta* (80.06 mm). However, the remaining subspecies did not display significant variation in this trait. Thus, differences in panicle length were detectable and can be utilized for taxonomic differentiation, yet it must be acknowledged that this is not universally applicable to all subspecies. Regarding the length of the first internode of the panicle, only *Hierochloë odorata* subsp. *baltica* (17.66 mm) exhibited a significant difference from the values observed in *Hierochloë hirta* subsp. *hirta* (23.80 mm) and *Hierochloë odorata* subsp. *odorata* (24.33 mm). Consequently, this trait appears less suitable for differentiation, although it reinforces the notion that variations exist among the subspecies of *Hierochloë odorata*, thereby supporting the taxonomic delineation of the species into subspecies.

No differences were observed in the length of the lateral branches originating from the panicle nodes, indicating that despite being analyzed, these parameters are not suitable for taxonomic differentiation based on the dataset.

Among the generative traits of Poaceae, the length of the spikelets plays a central role as a defining and consistent characteristic of the inflorescence structure. According to the present data on subspecies, *Hierochloë odorata* subsp. *odorata* possesses the shortest average spikelet length (4.73 mm), a value that falls within the size range (4.0–5.6(–6.3) mm) reported by Weimarck [50] in the original species description. Notably, spikelet length is not mentioned in the original species description or subsequent floristic works [63,64], which only include the length of the glumes without distinguishing between lower and upper glumes. This absence persists in later taxonomic descriptions as well [28,29]. Consequently, the present dataset provides entirely new information for taxonomic differentiation. The glume lengths reported by Weimarck [58] were thus compared to the spikelet length in this study. The values for *Hierochloë hirta* subsp. *praetermissa* (4.99 mm) did not significantly differ, reinforcing its classification within the *Hierochloë odorata* complex based on this trait. Conversely, *Hierochloë odorata* subsp. *baltica* exhibited the longest spikelets (5.534 mm) among the analyzed data, aligning with the size range ((4–)4.6–6.0(–8.0) mm) provided by Weimarck [54]. This subspecies did not show a significant difference from *Hierochloë hirta* subsp. *arctica*, which also had spikelets measuring 5.534 mm, a value reported within the original species description ((4.0–)4.5–6.0(–6.3) mm). Compared to the original species description, the present analysis contributes more effectively to the taxonomic classification of the subspecies.

The glume data, which were analyzed separately in this study by providing measurements for both the lower and upper glumes, offer new insights into taxonomic differentiation. A general analysis of the data indicates that, in all subspecies, the upper glumes were consistently longer than the lower ones. Based on the values of the lower glume, *Hierochloë odorata* subsp. *odorata* can be distinctly separated, as it exhibits the shortest average length (4.31 mm), significantly differing from the lower glume lengths of all examined subspecies.

The length of the outer lemma is a crucial diagnostic trait in flowers, a finding confirmed by this analysis, as it emerged as the second most effective generative trait, alongside spikelet length, in distinguishing subspecies. The results strongly support the clear separation of *Hierochloë odorata* subsp. *odorata*, which, with the shortest average outer lemma length (3.87 mm), is distinct from all other subspecies. The applicability of this trait is further demonstrated by the significant differentiation of *Hierochloë hirta* subsp. *praetermissa* (4.10 mm) and *Hierochloë odorata* subsp. *baltica* (4.79 mm) from all other subspecies.

Measurements and analyses were conducted for both the outer and inner lemmas, as well as for the awns of the outer lemmas in the first and second staminate flowers, providing a novel dataset for subspecies characterization. The awn of the outer lemma in the first flower emerges as a well-differentiated trait, allowing clear distinction between *Hierochloë odorata* and *Hierochloë hirta* as separate species, reinforcing the notion that maintaining these as distinct species is justified. Due to the shortness of the awn, three subspecies—*Hierochloë odorata* subsp. *odorata* (0.05 mm), *Hierochloë odorata* subsp. *baltica* (0.002 mm), and *Hierochloë hirta* subsp. *praetermissa* (0.025 mm)—form a distinct group. In contrast, *Hierochloë hirta* subsp. *hirta* is clearly separated due to its notably long awn (0.59 mm), which also aligns *Hierochloë hirta* subsp. *arctica* within the same group based on its similarly long awn (0.47 mm). This differentiation based on floral traits becomes even more pronounced in the second flower, where the greater awn length further reinforces these groupings.

The values of the inner lemma in both the first and second flowers further support the finding that *Hierochloë hirta* subsp. *praetermissa* (0.025 mm) does not differ from *Hierochloë odorata* subsp. *odorata*, whereas *Hierochloë odorata* subsp. *baltica* (4.09 mm) stands out as the only subspecies with a significantly distinct value.

*Hierochloë hirta* subsp. *praetermissa* was originally described under this name by G. Weim., as recorded by Weimarck, and this classification has been adopted by several authors [65,66,67]. However, Zvelev later classified *Hierochloë praetermissa* (G. Weim.) Prob. & Tzvelev as a distinct species [68], a perspective followed by other researchers [20,69]. Nevertheless, based on both the original description and the findings of the present study, generative traits do not strongly support this distinction. Wallnöfer [57] considered the taxon to be a subspecies of *Hierochloë odorata*, a classification that aligns with the present morphological evidence. Although it represents an intermediate form between *Hierochloë odorata* and *Hierochloë hirta*, as also noted by Weimarck [53], it is more appropriately assigned to *Hierochloë odorata*. The correct nomenclature, therefore, is *Hierochloë odorata* subsp. *praetermissa* G. Weim. ex B. Walln., a name that is internationally accepted and valid according to the International Plant Names Index, the Cite Taxon Page as ‘WFO,’ and is also listed in Plants of the World Online [11,70,71]. The present study further supports the adoption of this classification.

## 4. Materials and Methods

We examined herbarium specimens of *Hierochloë hirta* collected by Borbás at the Herbarium of the Hungarian Natural History Museum (BP). The characters of the specimens were measured at the Herbarium of Lund University (LD), focusing on those that were type specimens in Weimarck’s [53] monograph, except for the neotype of *Hierochloë hirta*. The neotype specimens of *Hierochloë hirta* [60] were examined at the Munich Botanical Garden Herbarium (M). The specimens of *Hierochloë odorata* subsp. *praetermissa* were also examined at the Lund Herbarium, which were not mentioned in Weimarck’s monograph. It is important to note that Weimarck [53] described it as *Hierochloë hirta* subsp. *praetermissa*, and Wallnöfer [58] changed its name to *Hierochloë odorata*, also assigning *Hierochloë hirta* to the *Hierochloë odorata* taxon. Additionally, we analyzed the original descriptions of *Hierochloë odorata* and *Hierochloë hirta* based on the works of Wahlenberg [72] and Schrank [73]. Several groups were included in the study and are presented below. The nomenclature of subspecies and taxa used in this study follows the works of Weimarck [53,57]. The final decision regarding the retention or modification of this classification was made at the conclusion of the study (herbarium data can be found in Appendix A and Appendix B):*Hierochloë odorata*○*Hierochloë odorata* (L.) Wahlenb. subsp. *odorata* G. Weim. (number of herbarium speciment: 9359, 9364 9395, 492192, 9365, 9363, 9361)○*Hierochloë odorata* (L.) Wahlenb. subsp. *baltica* G. Weim. (number of herbarium speciment: 486885, 486885, 9386, 9369, 9303)*Hierochloë hirta*○*Hierochloë hirta* (Schrank) Borbás subsp. *praetermissa* (G. Weim.) B. Walln. (number of herbarium speciment: 1221679, 1245158, 767323)○*Hierochloë hirta* (Schrank) Borbás subsp. *arctica* (Presl) G. Weim. (number of herbarium speciment: 9393, 522684, 9383, 9384)○*Hierochloë hirta* (Schrank) Borbás subsp. *hirta* G. Weim. (number of herbarium speciment: 9358, 570435, 505794, 487263, 487262)

Characteristics measured for each plant taxon (Figure 5):

1: length of panicle

2: length of first internode

3: 1st length of the longest panicle branch of the first node

4. 2nd length of the longest panicle branch of the first node

5: 1st length of the longest panicle branch of the second node

6: 2nd length of the longest panicle branch of the second node

7: 1st length of the longest panicle branch of the third node

8: 2nd length of the longest panicle branch of the third node

9: length of spikelet

10: length of the lower glume

11: length of upper glume

12: length of the lemma

13: length of the awn of the lemma of the first flower

14: length of palea

15: length of the lemma of second flower

16: length of the awn of the lemma of the second flower

17: length of palea the second flower.

6-6 spikelets from each inflorescence.

All morphological characters of the taxa were analyzed separately using non-parametric statistical methods. The Shapiro–Wilk test indicated that these variables did not follow a normal distribution (*p* < 0.05). Therefore, we applied the non-parametric Kruskal–Wallis test (α = 0.05). The non-parametric Dunn test with Bonferroni correction was used for multiple pairwise comparisons.

To evaluate the taxa and their morphological characters, we used the exploratory and unsupervised multivariate data analysis method, principal component analysis (PCA). This allows us to reduce the 17-dimensional space defined by morphological characters, so that the multidimensional mapping, and hence the distance between taxa, can be analyzed [74]. The differentiation of one subspecies from another can only be examined on a trait-by-trait basis. Principal component analysis (PCA) incorporates all measured traits, reducing the n-dimensional space defined by these traits into a plane determined by two principal components. This method allowed us to highlight the most significant distinguishing diagnostic traits among the subspecies. All data analysis was performed using XL-STAT statistical and data analysis solution software version 2024.4.1 [75].

## 5. Conclusions

The *Hierochloë* taxa are not considered to be in the genus *Anthoxantum*, which is in agreement with the opinion of the following authors. Pimentel et al. [25] delineated two sections within the *Anthoxanthum* genus. The *Anthoxanthum* section possesses a hermaphroditic terminal floret and two sterile florets, while the *Ataxia* section contains a hermaphroditic terminal floret and two lower florets that are mostly male or sterile, often with a palea. Both sections lack lodicules. In contrast, the *Hierochloë* genus is characterized by a terminal floret that is either hermaphroditic or female, accompanied by two male florets, each with lodicules. Lema-Suárez [26] clearly identified three distinct sections (Figure 6a–c). The two *Hierochloë* species analyzed in the present study, along with the spikelet floral structures, are visually presented by Conert [29], who detailed their spikelets (Figure 6d–g).

The panicle parameters of the examined taxa included traits previously described in species descriptions, but additional parameters not documented in these descriptions were also measured and evaluated. Statistical analysis of these traits using multiple methods revealed significant differences among the studied subspecies. However, multiple traits were required to confirm the taxonomic placement of taxa described as subspecies of a given species, as well as to clarify the classification of *Hierochloë hirta* subsp. *praetermissa* in several instances. This finding supports the suitability of these traits for species differentiation. The measurement data further confirmed that no single morphological trait allows for the complete separation of all subspecies. If such a trait existed, it would have appeared consistently in pairwise post hoc tests (a, b, c, d, e), yet this ideal scenario did not occur for any parameter. Instead, the most reliable traits were reflected in post hoc tests (a, b, c, d).

The analysis confirmed the distinction between the two species, *Hierochloë odorata* and *Hierochloë hirta*, demonstrating that taxonomic approaches grouping all taxa under *Hierochloë odorata* are not justified based on the present dataset. Three subspecies—*Hierochloë odorata* subsp. *odorata*, *Hierochloë odorata* subsp. *baltica*, and *Hierochloë hirta* subsp. *hirta*—exhibited differences across all panicle parameters and multiple individual traits. Although *Hierochloë hirta* subsp. *arctica* may not always be morphologically distinct, its geographic distribution and ploidy level provide sufficient justification for its recognition as a separate taxon under this name. The originally described *Hierochloë hirta* subsp. *praetermissa* exhibits affinities with all seven taxa examined, but it shows a stronger overlap with the typical *Hierochloë odorata* subsp. *odorata*, supporting its valid and justified designation as *Hierochloë odorata* subsp. *praetermissa* (G. Weim.) B. Walln.

## Figures and Tables

**Figure 1 plants-14-02270-f001:**
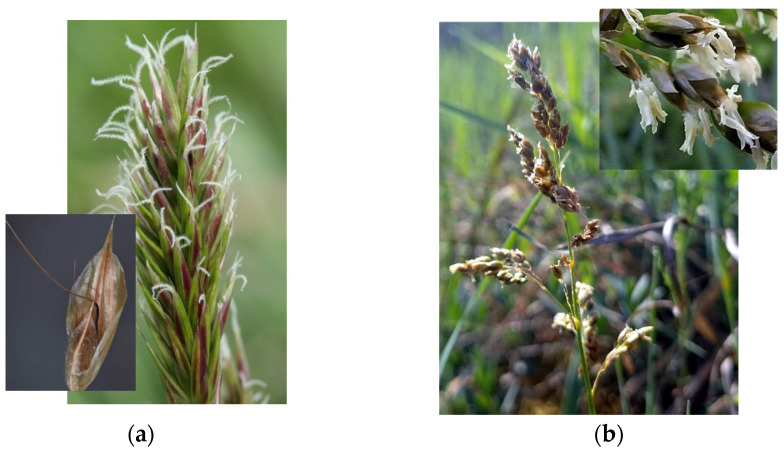
Comparison of the two genera. (**a**) The dense panicle inflorescence of *Anthoxanthum* and its spikelets; (**b**) the loose panicle of *Hierochloë* and its spikelets.

**Figure 2 plants-14-02270-f002:**
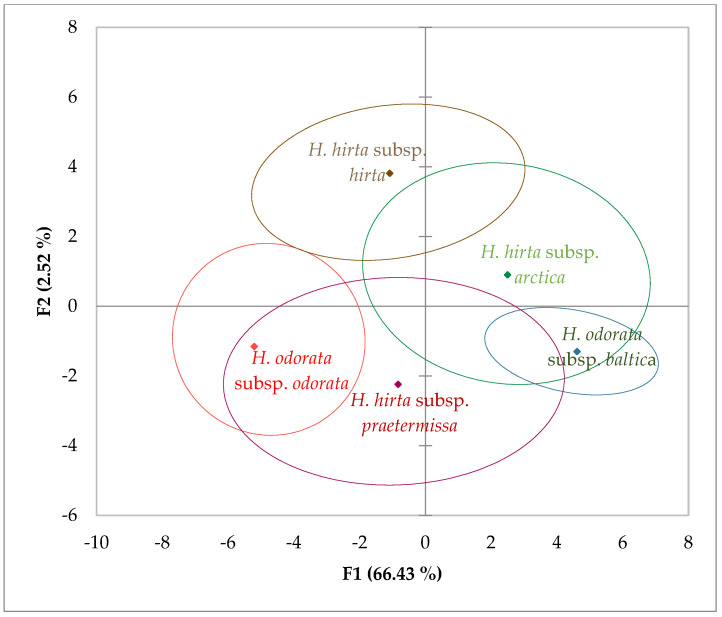
Observation chart and 95% confidence ellipses based on all measured inflorescence characters for each subspecies, with a cumulative variance of 96.13% for F1 and F2.

**Figure 3 plants-14-02270-f003:**
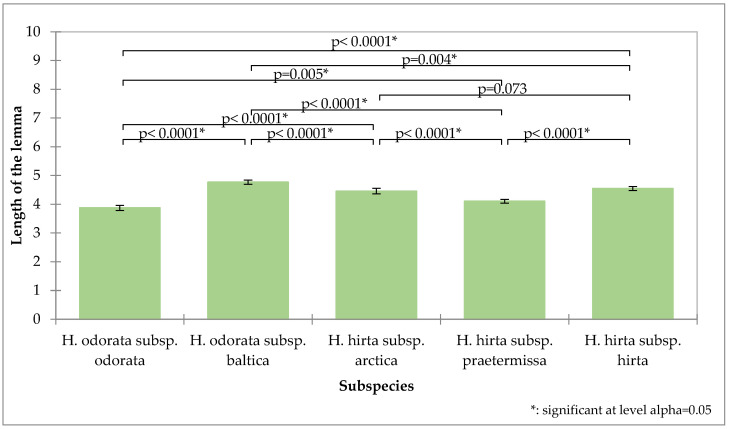
Analysis of the length of the first flower lemma of the tested subspecies, mean values, standard deviations, and probability values of the taxa analyzed (Kruskal–Wallis test, Dunn’s pairwise procedure with Bonferroni correction).

**Figure 4 plants-14-02270-f004:**
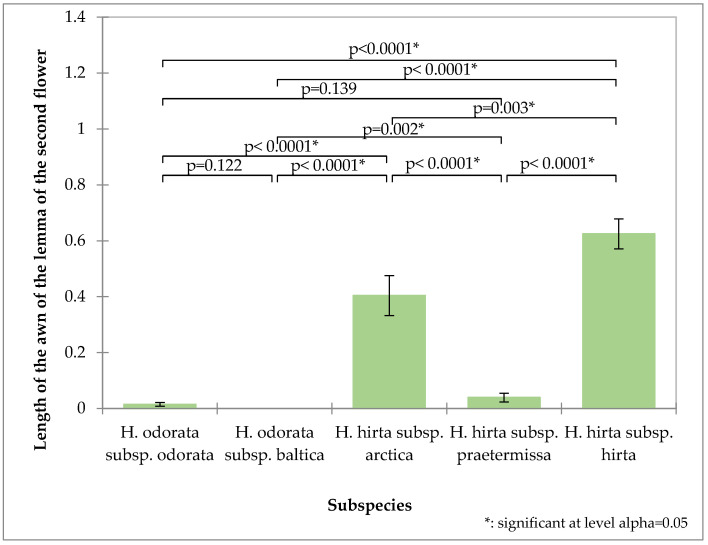
Analysis of the second flower awn of the tested subspecies: mean values, standard deviations, and probability values of the taxa analyzed (Kruskal–Wallis test, Dunn’s pairwise procedure with Bonferroni correction).

**Figure 5 plants-14-02270-f005:**
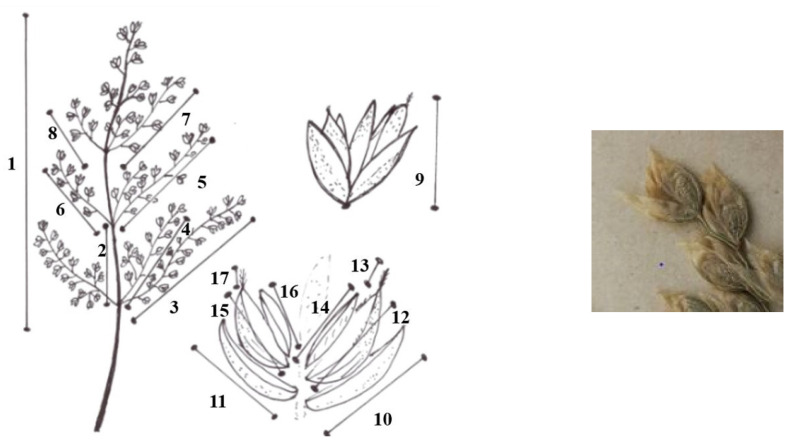
Characteristics measured (1: length of panicle; 2: length of first internode; 3: 1st length of the longest panicle branch of the first node; 4: 2nd length of the longest panicle branch of the first node; 5: 1st length of the longest panicle branch of the second node; 6: 2nd length of the longest panicle branch of the second node; 7: 1st length of the longest panicle branch of the third node; 8: 2nd length of the longest panicle branch of the third node; 9: length of spikelet; 10: length of the lower glume; 11: length of upper glume;12: length of the lemma; 13: length of the awn of the lemma of the first flower; 14: length of palea; 15: length of the lemma of second flower; 16: length of the awn of the lemma of the second flower; 17: length of palea the second flower).

**Figure 6 plants-14-02270-f006:**
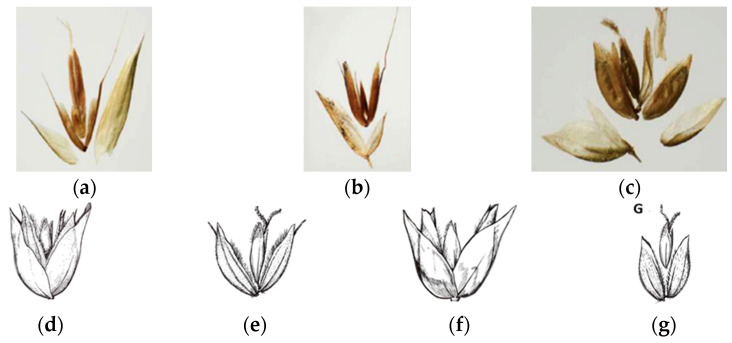
The three distinct sections. (**a**) *Anthoxanthum*; (**b**) *Ataxia*; (**c**) *Hierochloë* [2,26]; (**d**) spikelets of *Hierochloë hirta*; (**e**) flowers of *Hierochloë hirta* spikelets; (**f**) spikelets of *Hierochloë odorata*; (**g**) flowers of *Hierochloë odorata* spikelets [30].

**Table 1 plants-14-02270-t001:** Measured characters of the subspecies of *Hierochloë odorata* and *Hierochloë hirta.* Data are presented as mean ± standard deviation. Different letters represent significant difference between means within each column (*p* < 0.05; Kruskal–Wallis test, Dunn’s pairwise procedure with Bonferroni correction) (1: length of panicle; 2: length of first internode; 3: 1st length of the longest panicle branch of the first node; 4. 2nd length of the longest panicle branch of the first node; 5: 1st length of the longest panicle branch of the second node; 6: 2nd length of the longest panicle branch of the second node; 7: 1st length of the longest panicle branch of the third node; 8: 2nd length of the longest panicle branch of the third node; 9: length of spikelet; 10: length of the lower glume; 11: length of upper glume; 12: length of the lemma; 13: length of the awn of the lemma of the first flower; 14: length of palea; 15: length of the lemma of second flower; 16: length of the awn of the lemma of the second flower; 17: length of palea the second flower).

	*H. odorata* ssp. *odorata*	*H. odorata* ssp. *baltica*	*H. hirta* ssp. *arctica*	*H. hirta* ssp. *praetermissa*	*H. hirta* ssp. *hirta*
1	80.1 ± 12 a	59.8 ± 8.7 a	64.3 ± 15.0 ab	64.9 ± 12.2 abc	80.6 ± 23.1 bc
2	24.3 ± 5.0 b	17.7 ± 1.7 a	19.4 ± 4.0 ab	19.0 ± 3.6 ab	23.8 ± 7.3 b
3	40.0 ± 8.7 a	31.7 ± 3.4 a	34.1 ± 8.2 a	35.1 ± 6.3 a	36.5 ± 8.8 a
4	29.0 ± 6.5 a	25.2 ± 3.6 a	25.3 ± 6.1 a	27.4 ± 4.9 a	28.9 ± 6.5 a
5	28.4 ± 5.9 a	24.5 ± 3.2 a	25.5 ± 6.4 a	26.5 ± 3.6 a	28.0 ± 5.7 a
6	19.6 ± 4.8 a	18.3 ± 2.5 a	19.7 ± 5.3 a	18.4 ± 3.2 a	21.3 ± 3.1 a
7	22.9 ± 6.0 a	18.6 ± 2.9 a	18.9 ± 5.2 a	20.0 ± 5.3 a	20.8 ± 3.9 a
8	16.5 ± 4.1 a	14.1 ± 2.5 a	15.3 ± 3.6 a	15.9 ± 3.5 a	17.5 ± 3.8 a
9	4.7 ± 0.6 a	5.5 ± 0.5 c	5.5 ± 0.5 c	5.0 ± 0.5 ab	5.2 ± 0.5 b
10	4.3 ± 0.6 a	5.1 ± 0.4 c	5.0 ± 0.6 c	4.6 ± 0.5 b	4.7 ± 0.5 b
11	4.5 ± 0.6 a	5.4 ± 0.5 c	5.4 ± 0.5 c	4.9 ± 0.5 b	5.1 ± 0.5 b
12	3.9 ± 0.4 a	4.8 ± 0.3 d	4.5 ± 0.5 c	4.1 ± 0.3 b	4.5 ± 0.3 c
13	0.1 ± 0.1 b	0.0 ± 0.0 a	0.4 ± 0.3 c	0.0 ± 0.1 ab	0.6 ± 0.3 c
14	3.5 ± 0.4 a	4.1 ± 0.3 c	3.9 ± 0.4 b	3.6 ± 0.3 a	4.0 ± 0.3 b
15	3.7 ± 0.6 a	4.4 ± 0.3 c	4.2 ± 0.4 b	4.0 ± 0.4 a	4.4 ± 0.4 c
16	0.0 ± 0.0 ab	0.0 ± 0.0 a	0.4 ± 0.3 c	0.0 ± 0.1 b	0.6 ± 0.3 d
17	3.5 ± 0.4 a	4.2 ± 0.2 c	3.9 ± 0.4 b	3.6 ± 0.3 a	4.0 ± 0.3 b

## Data Availability

The data presented in this study are available on request from the corresponding author due to data being part of an ongoing investigation.

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
