# Peer review of "Evaluating Inflorescence Morphology in Two Species and Subspecies of the Genus Hierochloë R. Brown"

_plants, 2025, doi:10.3390/plants14152270_

Round 1
Reviewer 1 Report
Comments and Suggestions for Authors
The manuscript investigates the reliability of inflorescence morphological traits in herbarium specimens to differentiate closely related species and subspecies within the genus Hierochloë, specifically Hierochloë hirta and Hierochloë odorata and their European subspecies. Using type specimens and verified herbarium samples from multiple herbaria, the authors analyzed nine morphological characters from 460 spikelets through statistical methods including Kruskal–Wallis tests, Dunn’s pairwise comparisons, and principal component analysis (PCA). The study concludes that while some subspecies can be clearly distinguished based on inflorescence morphology, no single trait suffices for complete separation of all taxa. The authors recommend recognizing Hierochloë odorata subsp. praetermissa as a subspecies rather than a distinct species and affirm the validity of the species names H. hirta and H. odorata. They also call for further molecular studies to confirm subspecies distinctions.
-
The manuscript needs a clearer explanation or visualization of the morphological characters measured, possibly with more detailed figures for readers unfamiliar with grass morphology.
-
While the PCA results are discussed, more explicit mention of the variance explained by principal components and how this relates to taxonomic separation would strengthen the interpretation.
-
The study focuses only on European subspecies; the applicability of findings to other geographic populations or species within Hierochloë is not addressed. In the discussion section, it needs to discuss it with comparative species, please, from the literature https://doi.org/10.15407/ukrbotj81.04.259.
-
The manuscript does not explore potential environmental or preservation effects on morphological traits in herbarium specimens, which could affect reliability. You can see and cite these articles as well i.e., 10.3390/agronomy12051078
-
Some sections, especially the introduction, are dense with taxonomic history and could be streamlined for clarity and focus to the topic.
-
Include detailed illustrations or photographs of the key morphological traits analyzed to aid reader comprehension.
-
Discuss potential limitations related to herbarium specimen condition and how this might influence morphological measurements.
Author Response
Thank you for your comment. We have attached the reply.

Reviewer 2 Report
Comments and Suggestions for Authors
General comments:
The submitted work is exciting and contributes valuable information to the systematics of grasses. Despite a positive evaluation, several important elements require improvement.
Firstly, the title suggests an assessment of the usefulness of herbarium specimens for taxonomic research, but there's no reference to this issue in the text. It seems the title is a bit overstated – there's no discussion in the work regarding the usefulness of herbarium materials compared to fresh materials. Therefore, the otherwise valid approach to the problem using herbarium materials is an element of methodology, not research tactics. The methodology lacks a detailed description of the herbarium materials (this could be included in the publication's appendices), showing the origin of the herbarium specimens, their acquisition date, collection habitat, etc. This would significantly expand the information about the studied materials.
The abstract must be shortened. According to the publisher's guidelines, "The abstract should be a total of about 200 words maximum." It also needs to be formatted appropriately to include information regarding: "1) Background: Place the question addressed in a broad context and highlight the purpose of the study; 2) Methods: Describe briefly the main methods or treatments applied. Include any relevant preregistration numbers, and species and strains of any animals used; 3) Results: Summarize the article's main findings; and 4) Conclusion: Indicate the main conclusions or interpretations."
Tables – were all taxonomic units compared with each other or only as presented in the tables? In my opinion, there should be one common table. It can be formatted to fit on a single page. For example, feature names could be replaced with acronyms explained in the materials and methods section and below the table.
The letters indicating homogeneous groups within features are questionable. The text implies that comparisons were made between all taxa. The method of comparing means for individual features between taxa is completely unclear, as this is the only sensible approach. The description of tables 1 and 2 states that "Different letters represent significant difference between means within each column" – which would mean that mean values of features were compared with each other – which makes no sense. In this arrangement, comparisons can only be made within rows (the same feature for different taxa) and not columns.
The entire work requires language correction. The references also need to be corrected, as they have been edited quite carelessly.
Detailed Comments on the Text:
Use a period instead of a comma for decimal numbers.
Line # 4: "characteristic" not "caracteristic"
Lines # 335 – 340: Please verify once more because values given in the text do not refer to values in the tables.
Lines # 436 – 438: Please verify once more because values given in the text do not refer to values in the tables.
Lines # 570 – 571: Delete duplicated phrase: "1-8: morphological characters for the whole inflorescence; 9-17: 6-6 spikelets from each inflorescence."
Lines # 599 – 603: These sentences should be moved to Materials & Methods.
Line # 638: Correct: "Ogrodu Botanicznego KCRZG w Bydgoszczy"
Comments on the Quality of English Language
Language correction is mandatory for this publication.
Author Response

(The authors gave the same response as above.)

Round 2
Reviewer 2 Report
Comments and Suggestions for Authors
With the reviewer's valuable input fully integrated, this piece has not just improved, it has truly elevated its standing. Any ambiguities or areas of contention have been completely ironed out, and the breadth and depth of the material available to readers have been significantly enhanced. It unequivocally meets the highest standards and is now perfectly poised for publication.